# Molecular Recognition of Imidazole-Based Drug Molecules by Cobalt(III)- and Zinc(II)-Coproporphyrins in Aqueous Media

**DOI:** 10.3390/molecules28030964

**Published:** 2023-01-18

**Authors:** Galina Mamardashvili, Elena Kaigorodova, Ivan Lebedev, Nugzar Mamardashvili

**Affiliations:** G.A. Krestov Institute of Solution Chemistry of the Russian Academy of Sciences, Akademicheskay St. 1, Ivanovo 153045, Russia

**Keywords:** Co(III)-porphyrins, Zn(II)-porphyrins, imidazole, axial coordination, selective binding, molecular recognition, receptor

## Abstract

The methods of ^1^H NMR, spectrophotometric titration, mass spectrometry and elemental analysis are applied to determine the selective binding ability of Co(III)- and Zn(II)-coproporphyrins I towards a series of imidazole-based drug molecules with a wide spectrum of pharmacological activity (metronidazole, histamine, histidine, tinidazole, mercazolil, and pilocarpine) in phosphate buffer (pH 7.4) simulating the blood plasma environment. It is shown that in aqueous buffer media, Co(III)-coproporphyrin I, unlike Zn(II)-coproporphyrin I, binds two imidazole derivatives, and the stability of mono-axial Co-coproporphyrin imidazole complexes is two to three orders of magnitude higher than that of similar complexes of Zn-coproporphyrin I. The studied porphyrinates are found to have the highest binding ability to histamine and histidine due to the formation of two additional hydrogen bonds between the carboxyl groups of the porphyrinate side chains and the binding sites of the ligands in the case of histidine and a hydrogen bond between the amino group of the ligand and the carbonyl oxygen atom of the carboxyl group of the porphyrinate in the case of histamine. The structures of the resulting complexes are optimized by DFT quantum chemical calculations. The results of these studies may be of use in the design of biosensors, including those for the detection, control and verification of various veterinary drug residues in human food samples.

## 1. Introduction

Complexes of porphyrins with coordinationally unsaturated metal cations (MP, M = Fe(II, III), Co(II, III), Mn(II, III, IV), etc.) are widespread in the living nature surrounding us. Coproporphyrins (tetrapyrrole macroheterocycles, which have propionic acid residues in four β-pyrrole positions) are intermediates in the biosynthesis of many vital substrates. In particular, the level of copro- and uroporphyrins in the urine can be used to identify heme biosynthesis disorders in patients, which can be either congenital, as, for example, in case of hereditary fermentopathies, or acquired, as, for example, with liver diseases and hemolytic anemia [1,2,3]. At the same time, coproporphyrinates have the properties of stable photosensitizers, are characterized by low toxicity and can be easily eliminated from the body. These compounds can be used to treat arthrosis and arthritis and in photodynamic therapy of cancer and surface-localized infections [4,5,6,7].

The aim of this work was to study the processes of the molecular recognition of imidazole-containing drug molecules of different natures by complexes of coproporphyrin I with coordinationally unsaturated metal cations. Due to the electronic effects of substituents and structural features, imidazole and its derivatives exhibit a wide range of pharmacological and/or biological activities and are effectively used in the development of new broad-spectrum drugs, including those used for veterinary medicine. In this work, we studied the binding ability of Co(III)- and Zn(II)-coproporphyrinates I (CoCP and ZnCP, respectively) to antimicrobial and antiprotozoal metronidazole and tinidazole, antithyroid mercazolil, m-cholinomimetic pilocarpine, proteinogenic amino acid histidine, and histamine, one of the mediators that regulate the vital functions of the human body (Figure 1) [8]. 

The ability of metalloporphyrin molecules to selectively interact with substrates of different natures depends on the ability of the macrocycle and the substrate to form several intermolecular bonds with each other. For example, in addition to the donor–acceptor bond between the coordinationally unsaturated metal cation of the macrocycle and the electron donor atom of the ligand, there may be additional binding sites formed due to van der Waals and/or dispersion interactions, hydrogen bonds, π–π stacking, etc. [9,10,11]. For example, as [12,13] show, π–π interactions between the porphyrin receptor and the aromatic part of an amino acid significantly stabilize the Zn–porphyrin complex with tryptophan, in comparison with a similar complex with glycine (the stability constant of the complex increases from 60 L/mol in glycine to 1200 L/mol in tryptophan). In our previous work [14], we studied the recognition ability of two hydrophilic Co-tetraarylporphyrins with hydroxyl groups in the phenyl rings of the macrocycle towards imidazole and its derivatives (metronidazole and histidine) to find out whether additional hydrogen bonds could be formed between the hydroxyl groups of the receptor (porphyrinate) and the oxygen atoms of the nitro and carboxyl groups of the substrates. In the coproporphyrinates, the groups that are responsible for their selectivity in organic substrate binding are conformationally flexible propionic acid residues in four β-pyrrole positions of the macrocycle. The structures of the resulting complexes were optimized by DFT quantum chemical calculations. The calculated and experimental data were in good agreement with each other, confirming the ability of the coproporphyrinates to molecularly recognize imidazole-based drug molecules. 

In this work, zinc and cobalt were taken as metal ions coordinating the porphyrin ligand, since in buffer media both of these metals have the ability to additionally coordinate nitrogen-containing electron-donor molecules. Moreover, in the case of cobalt, it is possible to attach two ligand molecules with very high stability constants. The Zn(II)-porphyrin was used for comparison with the Co(III)-porphyrin to further demonstrate the differences in the behavior of these metals in aqueous and non-aqueous media.

## 2. Results and Discussion

The binding ability of ZnCP and CoCP(H_2_O)_2_ to imidazole (Im) and a number of imidazole-containing drug molecules (L1–L6) in phosphate buffer (pH 7.4) at 20 °C was studied by the methods of ^1^H NMR, spectrophotometric titration, mass spectrometry and elemental analyses.

### 2.1. Formation of Bis-Axial CoCP(H_2_O)_2_ Complexes with Imidazole Derivatives

Co(III) complexes of water-soluble tetraarylporphyrins (Co(III)P or CoP) are characterized by *bis*-axial coordination of both nitrogen- and oxygen-containing small organic molecules [15,16]. 

The Co 2p spectrum recorded from CoCP(Im) shows the main Co 2p3/2 photoemission line at 788.9 eV and its satellite at 793.8 eV. According to [17], this feature allows us to identify the Co ion in the compound as an intermediate spin (S = 1) Co(III) species. The X-ray photoelectron spectra of the CoCP(L) samples were studied using an X-ray photoelectron spectrometer Kratos Analytical AXIS Supra, with a high-power monochromatic Al Kα radiation source with a photon energy of 1486.6 eV (Appendix A).

Earlier, we studied the processes of complexation of Co(III)-tetraarylporphyrins (CoTCPP and CoTSPP) with various drugs based on nitrogen-containing heterocycles [18,19]. Table 1 and Figure 2a show data on the stability constants of CoTCPP axial complexes, taken in this work as the object of comparison, with a number of imidazole derivatives. The location of the carboxyl groups in this macrocycle (4-positions of the phenyl rings) makes it impossible for additional hydrogen bonds to be formed with these ligands. The stability constants of the mono- and diaxial CoTCPP complexes with different L differ insignificantly.

Like CoTCPP and CoTSPP, CoCP forms *bis*-aqua complexes in aqueous media at pH 7.4, and the replacement of the aqua-groups with organic bases represents a two-stage process:CoCP(H_2_O)_2_ + L ↔ CoCP(L)(H_2_O) (1a)
CoCP(L)(H_2_O) + L ↔ CoCP(L)_2_
(1b)

Each of the process stages is accompanied by its own characteristic spectral changes in the UV-Vis spectra of the reaction system. In particular, the replacement of the first water molecule with the ligand is characterized by a decrease in the intensity of the Soret band and its red shift by 3.5 nm (Figure 3a). The replacement of the second water molecule is characterized by less noticeable spectral changes (Figure 3b). 

Theoretical UV-Vis spectra were calculated using the ZINDO/S semi-empirical method. This method is adapted for transition element compounds and also helps in predicting and describing electronic transitions of metalloporphyrin molecules and related compounds (Appendix A).

It is known that the electronic transitions of porphyrins corresponding to the Soret band (B-band) in the visible region are carried out mainly due to the π→π* transitions between HOMO-1 and LUMO/LUMO + 1 orbitals.

Thus, two identical transitions (λ_1_ = 392.59, f = 1.98 and HOMO-1→ LUMO + 1, λ_2_ = 392.78 nm, f = 2.00) are observed in the symmetric di-aqua complex of cobalt coproporphyrinate (CoCP(H_2_O)_2_), due to the degeneracy in the energy of LUMO/LUMO + 1 orbitals. When the first imidazole molecule is coordinated (CoCP(Im)), the degeneracy from the frontier orbitals is partly removed and a lower-energy transition contributes to the Soret band (HOMO-1→ LUMO, λ = 394.31, f = 1.57), which is demonstrated as a red shift. Finally, when two imidazole molecules coordinate (CoCP-(Im)_2_), the vacant boundary orbitals become degenerate again, and two identical electronic transitions are realized (λ_1_ = 396.45 nm, f = 1.70 and λ_2_ = 396.49 nm, f = 1.69).

The results of quantum chemical calculations are in good agreement with the experimental red shift of the Soret band during complexation. Appendix A shows the calculated electronic spectra for the complexes, as well as the scheme of electronic transitions.

The calculated constants of the water molecules’ replacement for the imidazole compounds (K^1^ and K^2^) are listed in Table 1 and in Figure 2c,d. As the presented data show, the CoCP binding ability towards imidazole in aqueous media is lower than that of hydrophilic CoTCPP. Moreover, the stability of the mono- and diaxial Co(III)-coproporphyrin complexes with the imidazole derivatives (L1–L6) strongly depends on the presence of additional binding sites in the latter (Figure 2).

The obtained dependence of the stability constants on the ligand nature and the calculated thermodynamic parameters of the axial coordination process, which will be discussed below, suggest that at least four of the six studied drug molecules, when coordinated on the macrocycle zinc cation, form additional hydrogen bonds with the binding sites of the conformationally mobile propionic acid branches of CoCP. The most stable complexes are formed with histidine and histamine, probably due to the formation of fairly strong hydrogen bonds between the porphyrinate binding sites and amine and/or carboxyl groups of the ligand (6.5 and 5.8 kJ/mol, respectively). The stability constant of the CoCP(L2) and CoCP(L3) complexes is higher than that of the CoCP(Im) complex by an order of magnitude.

In the Co(III)-5,15-bis-(4-sylfophenyl)-10,20-bis-(3-hydroxyphenyl)-porphyrin and Co(III)-5,15-bis-(4-sylfophenyl)-10,20-bis-(2-hydroxyphenyl)-porphyrin described by us in [14], the selectivity of binding imidazole derivatives is only observed when the first ligand is bound. When the second axial ligand is attached, the binding process is not selective. This is probably explained by the fact that in the case of these porphyrins, the second complexation step is only due to the single-center binding of the ligands through the metal cation of the macrocycle (i.e., the sulfo- and hydroxy-binding sites rigidly fixed on the macrocycle periphery are not capable of forming additional bonds with the ligand molecule).

At the same time, it was established that the CoCP’s ability to molecularly recognize imidazoles is also retained upon coordination of the second ligand, although the energy of additional binding for the second ligand is somewhat lower. This, apparently, means that the coordination of the second imidazole derivative is also accompanied by the formation of additional binding centers.

### 2.2. ZnCP Mono-Axial Complexes with Imidazole Derivatives

In both non-aqueous and aqueous media, the zinc cation in the composition of porphyrin macrocycles is in the oxidation state (II) and forms mono-axial ZnCP(L) complexes with strong organic bases, such as imidazole and its derivatives [20,21,22]. The data presented in Figure 4 indicate that complexation of Zn-porphyrin with the organic ligand results in the formation of 1:1 complexes (there is one family of spectral curves in the UV-Vis spectrum of the reaction system, Figure 4a, and there is one step on the respective curves of spectrophotometric titration, Figure 4b). The shift in the Soret band that accompanies the formation of the ZnCP mono-ligand complex is ~7 nm.

The stability constants of the mono-axial ZnCP complexes with various imidazole-containing drugs and the enthalpies of their formation are shown in the diagram (Figure 5) and in Table 1.

As the presented data show, ZnCP’s binding ability to imidazole in aqueous media is three to four times lower than that of hydrophilic Zn-tetraarylporphyrins. Moreover, the stability of the *mono*-axial ZnCP complexes with the imidazole derivatives (L1–L6) also strongly depends on the presence of additional binding sites in the latter (Figure 5). The ZnCP(L) stability constants for the imidazole derivatives have a similar relationship to those of CoCP(L) and CoCP(L)_2_. This means that upon coordination of the ligands on ZnCP, the studied substrates form similar additional bonds with the propionic acid fragments of the porphyrinate. The most stable complexes are formed with histidine and histamine, probably due to the formation of fairly strong hydrogen bonds between the porphyrinate binding sites and amine and/or carboxyl groups of the ligand (6.5 and 5.8 kJ·mol^−1^, respectively). The stability constant of the ZnCP(L2) and ZnCP(L3) complexes is higher than that of the ZnCP(Im) complex by more than an order of magnitude.

### 2.3. Geometry Optimization of ZnCP(L) and CoCP(L)

Table 2 shows the calculated energy and geometric parameters of ZnCP complexes with axial ligands (L1–L6). An NBO analysis was performed to predict the natural molecular charge distribution as well as to calculate the stabilization energies between the ligand and the porphyrinate zinc cation. The interaction between the donor and acceptor orbitals was analyzed based on the second-order perturbation stabilization energy (E_st_) and the charge transfer value (*q*_CT_) presented in Table 2. The intermolecular interaction energy of the studied complexes with the axial ligands was also calculated by taking into account the BSSE superposition error (−E_int_).

The NBO analysis shows that the Zn–N bond in the coordination center was identical for all of the calculated structures and was formed by the partial transfer of the nitrogen atom electron pair to the free orbitals of the zinc cation (LP N → LP * Zn). The interaction between the donor and acceptor orbitals was analyzed based on the second-order perturbation stabilization energy. The E_st_ value increased as the interaction energy between the host and guest molecules became higher. The stabilization energy between the central zinc cation and the nitrogen atom of the macrocycle pyrrole ring, according to the NBO analysis data, was 43–49 kcal·mol^−1^, which is higher than the stabilization energy of Zn-porphyrinate with axial ligands.

The calculation results show that the interaction of the macrocycle zinc cation with the nitrogen atom of the amino acid led to the elongation of the Zn(II) ion from the macrocycle plane. This can be seen from the values of the N–Zn–N angle, which decreased proportionally to the increase in the energy of the axial ligand binding to the Zn-porphyrin (Table 2). At the same time, the stabilization energy of the Zn–N bond decreased in inverse proportion to the energy of the porphyrinate interaction with amino acids, which was in the range of 11–20 kcal/mol. Thus, the highest binding energy of Zn-porphyrin was observed in the case of histamine (L2) and decreased in the series L2 > L3 > L6 > L4 > L5 > L1.

As the analysis of the optimized structures shows, the higher binding energy of L2 and L3 is probably associated with the formation of additional hydrogen bonds between the porphyrinate and the ligand. The lengths of these bonds are 1.9 and 2.0 Å, respectively, which allows them to be classified as medium-strength hydrogen bonds.

Similar geometric, electronic and energy parameters calculated for the CoCP(L) *mono*-ligand complexes are presented in Table 3. The geometric optimization of the extra-complexes was carried out by taking into account the above-described features of the geometric structure of the initial molecules. During the optimization, many different configurations were taken into account, differing in the presence and type of hydrogen bonds between the periphery of the macrocycle and extra-ligand. Figure 6 shows the optimized structure of the complexes corresponding to configurations with the minimal energy.

The calculated Zn–N and Co–N bond lengths and the N–Zn–N and N–Co–N angles given in Table 2 and Table 3 are in good agreement with the literature data [23,24,25,26]. 

It is obvious that a significant contribution to the energy of the intermolecular interaction between the extra-ligand and porphyrinate is provided by hydrogen bond formation, which is observed in all of the considered *extra*-complexes.

According to Table 4, the strongest hydrogen bonds are observed in the CoCP(L2) and CoCP(L3) extra-complexes. In the first case, the significant energy gain is due to the formation of a cyclic hydrogen bond between the carboxyl groups, as it occurs in carboxylic acid dimers, and in the second, an abnormally strong hydrogen bond is formed between the amino group of the extra-ligand and the carboxyl group on the macrocycle periphery. The high stabilization energy can also indicate proton transfer processes in the CoCP(L3) complex, similar to the proton transfer in amino acids (Figure 7).

In conclusion, it should be said that the calculated values of the intramolecular interaction energy for the *extra*-complexes in free state are consistent with the thermodynamic stability constant of the complexes calculated from the experimental data (Figure 8 and Figure 9). However, it should be taken into account that the real systems have been studied in buffer solutions, the components of which, together with the solvent, can also participate in the interactions.

### 2.4. ^1^H NMR Studies

The formation of the MCP(L) metal complexes was readily observed by ^1^H NMR spectroscopy. The characteristics of the ^1^H NMR spectra of a series of CoCP(L), which can be formed without an excess of a substrate, are presented in Table 5 and Figure 10.

The axial coordination of the L1–L6 ligands on the Co(III) cation of the macrocycle is evidenced by the characteristic high-field shifts of the signals of the ligand imidazole fragment, which are in the immediate vicinity of the ring current of the aromatic tetrapyrrole macrocycle.

The additional hydrogen bonds in the MCP(L) formed between the O-H, N-H, and S-H groups of the macrocycle and ligand are predominantly of an electrostatic nature. Their formation is always accompanied by a strong downfield shift in the resonant signal involved in proton binding [27,28,29]. D6-DMSO is a solvent capable of forming very strong hydrogen bonds with the above protons (Table 5). The signals of such protons in d6-DMSO are narrower and shifted downfield compared to the other solvents.

During the formation of axial CoCP(L) complexes with additional hydrogen bonds (Table 4), which are stronger than H···d6-DMSO, the signals are also shifted downfield compared to the signal positions in the L1–L6 spectra. In the ^1^H NMR spectra of CoCP(L1–L4, L6), the proton signal of the porphyrinate carboxyl groups (broadened singlet) is split and partially shifted downfield (Figure 10). Similar downfield shifts are observed in the proton signals of the CoCP(L1) hydroxy group, the CoCP(L3) amino group, and the CoP(L5) HS group (Table 5). The fact that the -COOH resonances of the 1:1 CoCP and L2 mixture appear as four sharp signals, which are significantly shifted downfield from their original position in the individual components (CoCP and L2), confirms that two OH···O hydrogen bonds are formed in the CoP(L5) complex.

The existence of the most durable mono-axial complexes was also recorded by mass spectrometry (Figure 11). In the mass spectra of the CoCP(L2) and CoCP(L3) complexes obtained in the negative shooting mode, the peaks of the target ions [M]^−^ (*m*/*z* 835.7) and (*m*/*z* 821.8), respectively, were recorded. The highest intensity peak in the mass spectra of both complexes was formed as a result of the elimination of the axial ligands and corresponds to the CoCP ion (*m*/*z* 710.7).

Appendix A shows the powder diffraction patterns of CoCP, one of the studied ligands (L2), and the CoCP(L2)2 complex. The diffraction pattern of the CoCP(L2)2 complex differs from the diffraction pattern of the starting compounds. This sample is less crystalline (more amorphous), which does not allow any correspondence with our simulated samples.

## 3. Materials and Methods

### 3.1. Compounds

Coproporphyrin I (3,8,13,18-tetramethyl-2,7,12,17-tetrapropionic acid-21,23-porphyrin), ZnCl_2_, CoCl_2_, imidazole, 2-(2-methyl-5-nitro-1H-imidazol-1-yl)ethanol (metronidazole, L1), 2-amino-3-(1H-imidazol-4-yl)propionic acid (histidine, L2), 4-(2-aminoethyl)imidazole (2-(1H-imidazol-4-yl)ethanamine)(histamine, L3), 1-(2-ethylsulfonylethyl)-2-methyl-5-nitroimidazole (L4), 1-methyl-2-imidazolethiol (L5), and hydrochloride of (3S-cis)-3-ethyldihydro-4-[(1-methyl-1H-imidazol-5-yl)methyl]-2(3H)-furanone(pilocar-pine, L6) were analytical grade reagents supplied by Sigma-Aldrich and Merck. Ethanol was purchased from Fluka.

### 3.2. Synthesis and Characterization

#### 3.2.1. Synthesis

Zn(II)-coproporphyrin I and Co(III)-coproporphyrin I were obtained from commercially available coproporphyrin I by means of the method described in [30]. The product of the interaction of coproporphyrin I with cobalt chloride in ethanol in a nitrogen atmosphere is Co(II)-coproporphyrin. When planted in water in an air atmosphere, Co(II)-coproporphyrin is oxidized to diaqua-Co(III)-coproporphyrin.

**Zn(II)-coproporphyrin I (ZnCP)**. Elemental analysis: calcd. (%) for C_36_H_36_N_4_O_8_Zn·H_2_O: C, 58.74; H, 5.20; N, 7.61; found: C, 58.51; H, 5.18; N, 7.56. ^1^H NMR (500 MHz, d6-DMSO): δ, 12.28 (s, 4H); 10.27 (s, 4H), 4.28 (t, 8H), 3.87 (t, 12H), 3.22 (m, 8H). ESI-Mass: MS (API-ES): (M = 718.04) neg, (*m*/*z*): 717.9, 653.4, 479.4. UV-Vis spectrum in buffer (pH 7.4), λ_max_, nm (lgε): 403.4 (5.15), 535.3 (3.62), 571.2 (3.95).

**Co(II)-coproporphyrin I (Co(II)CP)**.Elemental analysis: calcd. (%) for C_36_H_36_N_4_O_8_Co: C, 60.76; H, 5.10; N, 7.87; found: C, 60.58; H, 5.07; N, 7.81. ^1^H NMR (500 MHz, d6-DMSO): δ, 15.47 (br.s, 4H), 4.68 (br.s, 8H), 4.45 (br.s, 12H), 4.08 (br.s, 8H). ESI-Mass: MS (API-ES): (M = 711.62) neg, (*m*/*z*): 710.9. UV-Vis spectrum in DMF, λ_max_, nm (lgε): 392.4 (5.15), 529.3 (3.62), 562.2 (3.95).

**Co(III)-coproporphyrin I [CoCP or CoCP(H_2_O)_2_]**. Elemental analysis: calcd. (%) for C_36_H_36_N_4_O_8_Co·H_2_O: C, 59.26; H, 5.25; N, 7.68; found: C, 58.98; H, 5.21; N, 7.54. ^1^H NMR (500 MHz, d6-DMSO): δ, 12.20 (s, 4H), 10.24 (s, 4H), 4.25 (m, 8H), 3.72 (t, 12H), 3.16 (m, 8H). ESI-Mass: MS (API-ES): (M = 711.62) neg, (*m*/*z*): 710.7. UV-Vis spectrum in buffer (pH 7.4), λ_max_, nm (lgε): 405.2(5.12), 536.8 (3.59), 573.0 (3.9).

**Co(III)-tetra(4-carboxyphenyl)porphyrin [CoTCPP or CoTCPP(H_2_O)_2_]** was prepared as described in [16]. Elemental analysis: calcd. (%) for C_48_H_28_CoN_4_O_8_: C, 68.01; H, 3.33; N, 6.61; found: C, 67.80; H, 3.31; N, 6.57. ^1^H NMR (500 MHz (d6-DMSO): δ, 9.22 (s, 8H), 9.05 (m, 8H), 8.26 (m, 8H). ESI-Mass: MS (API-ES): (M = 847.7) neg, (*m*/*z*): 846.5. UV-vis spectrum in buffer (pH 7.4), λ_max_ nm, (logε): 426.7(5.01), 540.0 (3.96).

#### 3.2.2. Characterization

**Spectrophotometric studies.** The UV-Vis spectra were recorded at 25 °C on a Jasco V-770 spectrometer. The thermodynamic constants for the complexation of CoCP and ZnCP with a ligand (L) were calculated by the formula based on the spectrophotometric titration experiment results:K=[MP−L][MP]⋅[L]=1[L](ΔAi,λ1ΔAo,λ1⋅ΔAo,λ2ΔAi,λ2) ,M−1
or
K1=[MP−L][MP]⋅[L]=1[L](ΔAi,λ1ΔAo,λ1⋅ΔAo,λ2ΔAi,λ2) ,M−1
K2=[MP−2L][MP−L]⋅[L]=1[L](ΔAi,λ1ΔAo,λ1⋅ΔAo,λ2ΔAi,λ2) ,M−1
where λ_1_ is the descending wavelength; λ_2_ is the ascending wavelength; [MP-L] is the concentration of the porphyrinate with one axial ligand; and [L] is the ligand concentration. Δ*A*_o_ is the maximal change in the solution optical density at the given wavelength, and Δ*A_i_* is the change in the solution optical density at the given wavelength at the given concentration [31].

**NMR studies.** All the NMR experiments were performed on a Bruker Avance III-500 NMR spectrometer equipped with a 5 mm probe using standard Bruker TOPSPIN Software. TMS signals were used as the internal standards to determine the chemical shifts. The temperature control was performed using a Bruker variable temperature unit (BVT-2000) in combination with a Bruker cooling unit (BCU-05) to provide chilled air. The experiments were performed at 298 K without sample spinning.

**The elemental analyses** were performed on a Flash EA 1112 C, H, N analyzer. 

**The mass-spectra were** obtained on a Shimadzu Biotech Axima Confidence MaldiTof mass-spectrometer (with dihydroxybenzoic acid as the matrix).

**The X-ray photoelectron spectra** were studied on an X-ray photoelectron spectrometer Kratos Analytical AXIS Supra, with a high-power monochromatic Al Kα radiation source with a photon energy of 1486.6 eV. 

**X-ray powder diffraction (XRPD)** measurements were carried out on a Bruker AXS D8-Advance diffractometer (Bruker AXS).

#### 3.2.3. Details of Quantum Chemical Calculations

The density functional theory (DFT) calculations were carried out using the GAUSSIAN 16 quantum chemical program package [32]. The geometry optimization, vibrational frequency calculations and conformational analysis of the complexes and individual molecules in the ground state were performed using the B97D3 functional [33,34] in combination with the 6-31 + G* [35] basis set. The analysis of the electron density distribution was carried out using the NBO program [36] implemented in GAUSSIAN 16. All of the calculations were performed for isolated molecules in vacuum, without taking into account the effect of the solvent. The results of the quantum chemical calculations were visualized with the ChemCraft program [37].

## 4. Conclusions

Thus, this study has shown that Co(III) complexes of coproporphyrin I have the ability to bind one or two molecules of imidazole-containing drugs with a wide range of pharmacological activities (metronidazole, histamine, histidine, tinidazole, mercazolil, and pilocarpine) in phosphate buffer (pH 7.4), simulating the blood plasma environment. The stability of the resulting mono- and diaxial porphyrin complexes with the ligands depends on whether the latter has a nitrogen-containing heterocycle (imidazole) and various binding sites capable of forming additional hydrogen bonds with conformationally flexible propionic acid side substituents of the macrocycle (there are two hydrogen bonds between the carboxyl groups in the case of histidine, and a hydrogen bond between the amino group of the ligand and the carbonyl oxygen atom of the carboxyl group of the porphyrinate in the case of histamine). The spectral responses to the substrate binding processes were identified and characterized, and the stability constants of the resulting complexes and concentration ranges of their existence were determined. The results of these studies may be of use in the design of biosensors, including those for the detection, control and verification of various veterinary drug residues in human food samples.

## Figures and Tables

**Figure 1 molecules-28-00964-f001:**
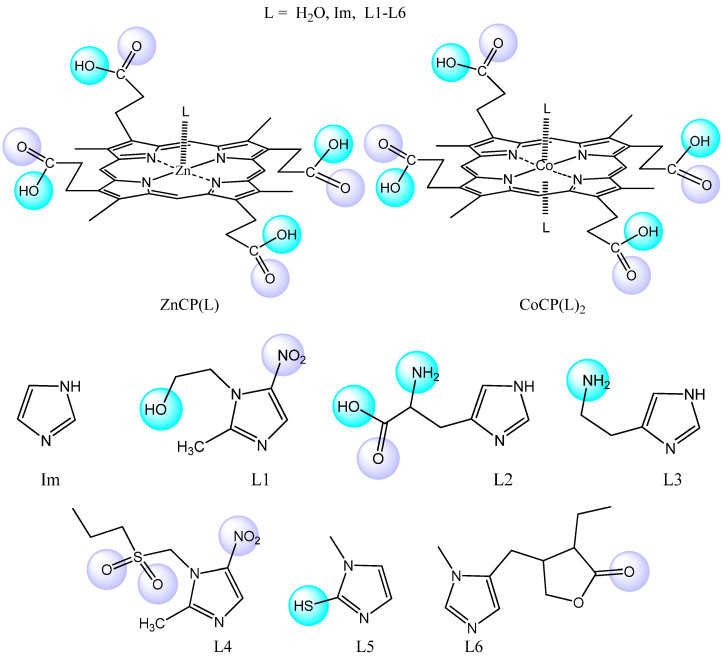
Structures of the studied compounds.

**Figure 2 molecules-28-00964-f002:**
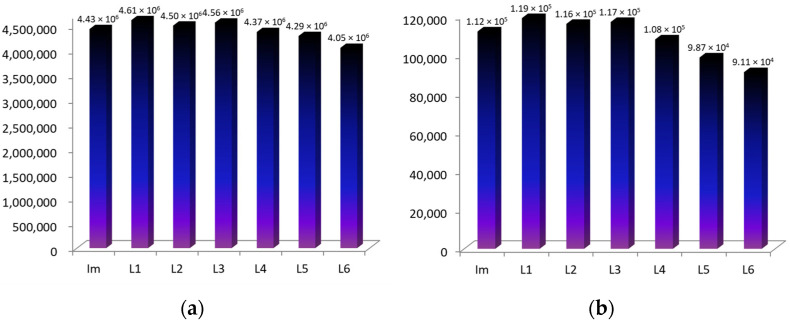
Equilibrium constants K^1^, M^−1^ (**a**,**c**) and K^2^, M^−1^ (**b**,**d**) for the axial ligand exchange reactions of water-soluble CoTCPP (**a**,**b**) and CoCP (**c**,**d**) with imidazole-containing organic bases (the estimated uncertainty in the *K* values is ± 5%).

**Figure 3 molecules-28-00964-f003:**
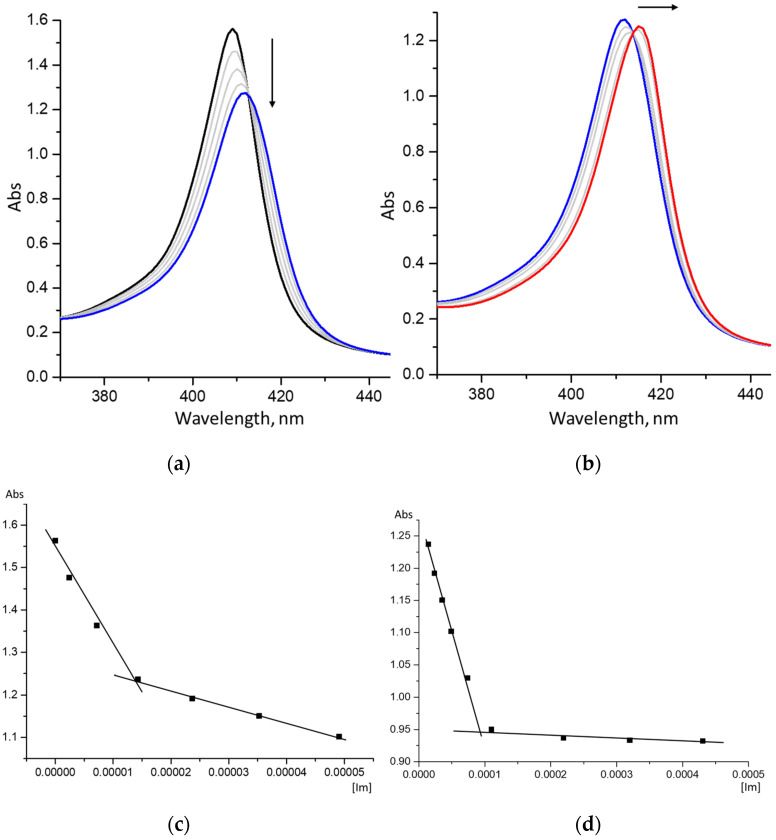
Data of spectrophotometric Im-titration of CoCP(H_2_O)_2_: (**a**) the first titration step and titration curve (**c**), corresponding to the formation of CoCP(Im) (C_porph_. = 8 × 10^−6^ mol/L; C_Im_ = 0 ÷ 8 × 10^−6^ mol/L); (**b**) the second titration step and titration curve (**d**), corresponding to the formation of CoCP(Im)_2_ (C_porph_. = 8 × 10^−6^ mol/L; C_Im_ = 8 × 10^−6^ ÷ 2 × 10^−4^ mol/L).

**Figure 4 molecules-28-00964-f004:**
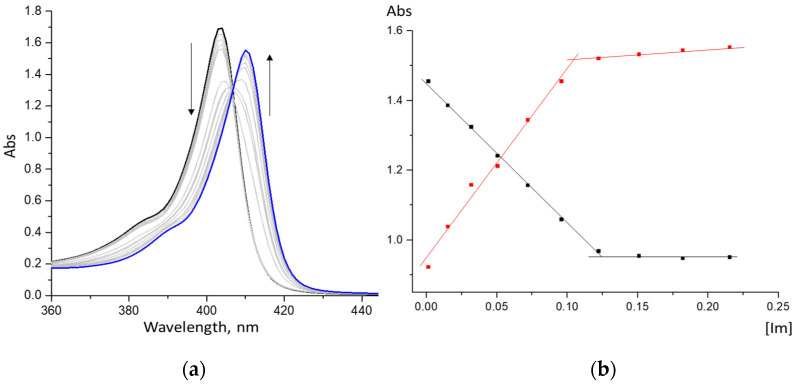
Spectrophotometric Im-titration of ZnCP in phosphate buffer (pH 7.4): (**a**) changes in UV-Vis spectra; (**b**) curves of spectrophotometric titration at two wavelengths—“decreasing” at 400 nm and “increasing” at 412 nm.

**Figure 5 molecules-28-00964-f005:**
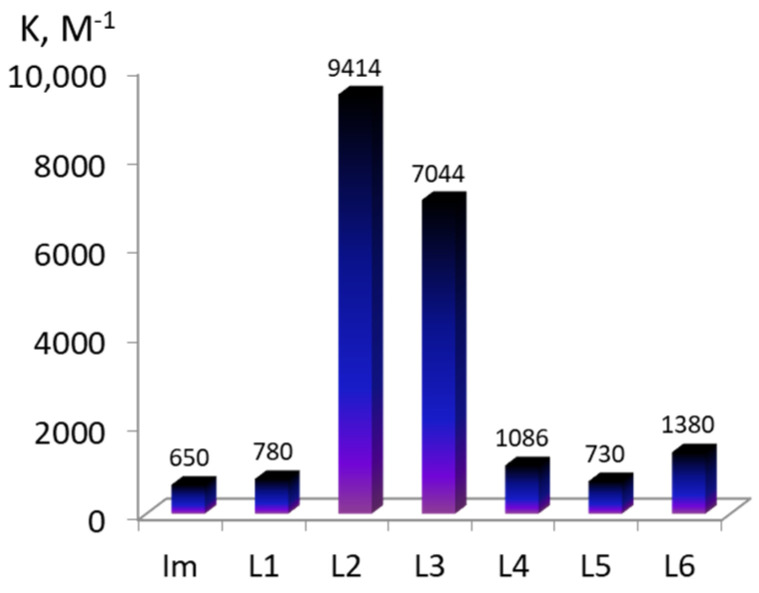
Equilibrium constants (*K,* M^−1^) for the axial ligand exchange reactions of water-soluble ZnCP with nitrogen-containing organic bases (the estimated uncertainty in the *K* values is ± 5%).

**Figure 6 molecules-28-00964-f006:**
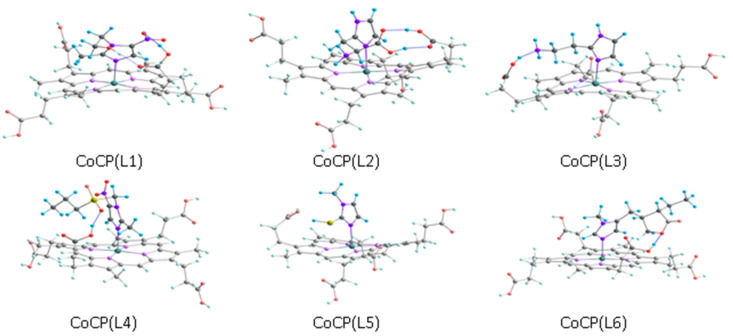
Optimized structures of the extra-complexes corresponding to configurations with minimal energy.

**Figure 7 molecules-28-00964-f007:**
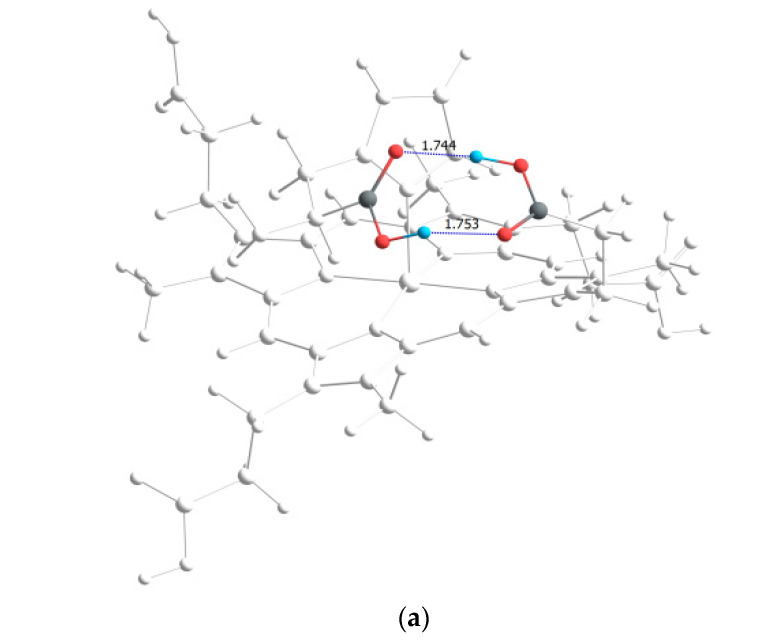
The strongest intramolecular hydrogen bonds in CoCP(L2) (**a**) and CoCP(L3) (**b**).

**Figure 8 molecules-28-00964-f008:**
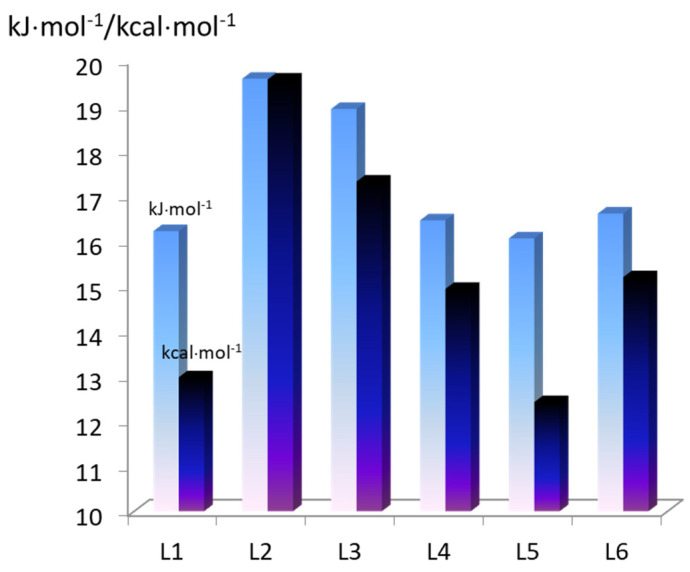
Intermolecular interaction energies of the studied complexes with axial ligands, taking into account the BSSE superposition error (−E_int_) (kcal·mol^−1^) (the blue bar) and the total Gibbs energies of ZnP(L) complex formation (−ΔG^g^, kJ·mol^−1^), calculated from the experimental data (the cyan bar).

**Figure 9 molecules-28-00964-f009:**
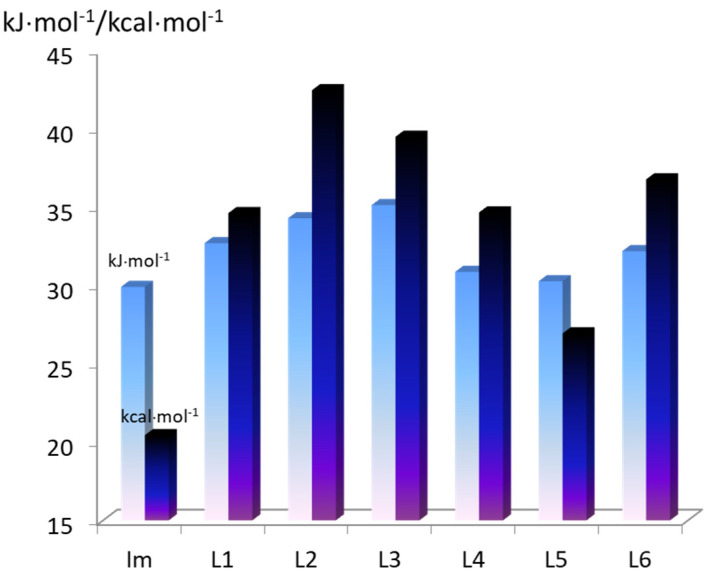
Intermolecular interaction energies of the studied complexes with axial ligands, taking into account the BSSE superposition error (−E_int_) (the blue bar) and the total Gibbs energies of CoP(L) complex formation, calculated from the experimental data (the cyan bar).

**Figure 10 molecules-28-00964-f010:**
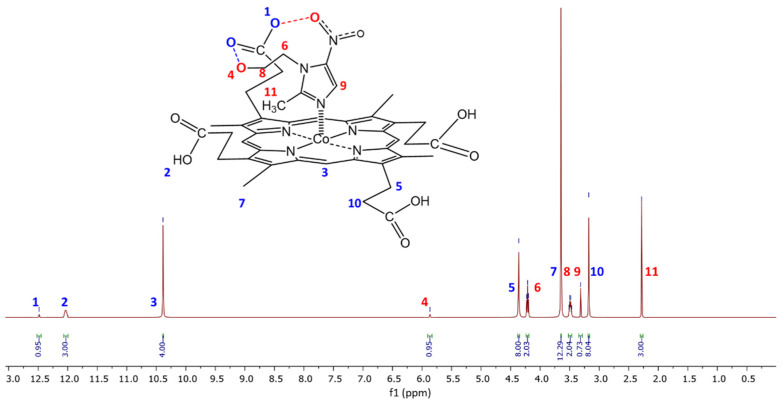
^1^H NMR spectrum of CoCP(L1) in d6-DMSO at 25 °C.

**Figure 11 molecules-28-00964-f011:**
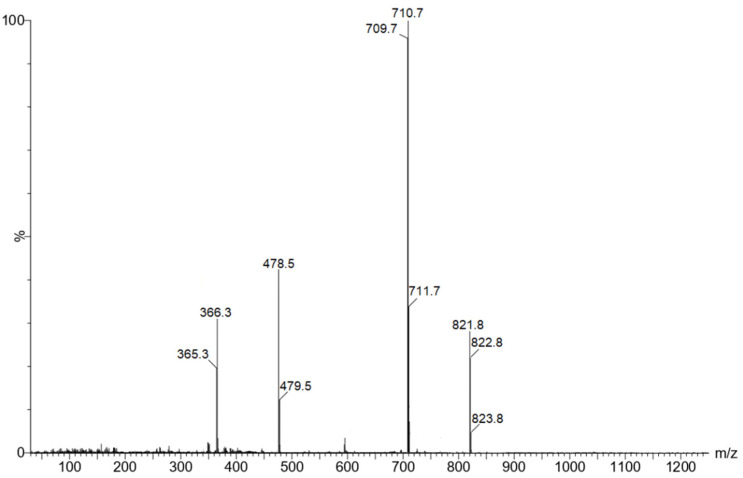
Mass spectrum of CoCP(L3).

**Table 1 molecules-28-00964-t001:** Total energies of formation of ZnCP(L), CoCP(L) and CoCP(L)_2_ (−ΔG^g^, kJ·mol^−1^), M-N(Im) bond energies (−ΔG^M-L^, kJ·mol^−1^) and hydrogen bond energies (−ΔG^HB^, kJ·mol^−1^).

	CoTCPP + L		CoTCPP(L) + L		CoCP + L		CoCP(L) + L		ZnCP + L	
	−ΔG^g a^	−ΔΔG^M-N b^	−ΔG^g^	−ΔΔG^M-N^	−ΔG^g^	−ΔΔG^MN+HB c^	−ΔG^g^	−ΔΔG^MN+HB^	−ΔG^g^	−ΔΔG^MN+HB^
**Im**	37.26	0	28.31	0	29.90	0	25.02	0	15.77	0
**L1**	37.36	0.10	28.46	0.15	32.69	2.27	26.44	1.42	16.21	0.44
**L2**	37.30	0.04	28.40	0.09	34.28	4.38	28.13	3.11	22.29	6.52
**L3**	37.33	0.07	28.41	0.10	35.12	5.22	28.54	3.52	21.58	5.81
**L4**	37.23	−0.03	28.22	−0.09	30.86	0.96	26.14	1.12	17.02	1.25
**L5**	37.18	−0.08	28.00	−0.31	30.26	0.36	25.61	0.59	16.05	0.28
**L6**	37.04	−0.22	27.81	−0.50	32.18	2.28	26.19	1.17	17.60	1.83

^a^ The total energy of MP(L) or MP(L)_2_ formation (ΔG = −RT lnK, T = 293 K); ^b^ the relative energy of MP(L) or MP(L)_2_ formation due to different M–L bond strengths; ^c^ the relative energy of the formation of MP(L) or MP(L)_2_ complexes due to different M–L bond strengths (<0.5 kJ·mol^−1^) and hydrogen bonding in the complex (>0.5 kJ·mol^−1^).

**Table 2 molecules-28-00964-t002:** Energy and geometric parameters of ZnCP(L) in the gas phase.

Ligands	−E_int_ kcal·mol^−1^	Zn-L, Å	*E*_st_(Zn-L) kcal·mol^−1^	*q*_CT_ (L), *e*	*E*_st_(Zn-Np) kcal·mol^−1^	*q*_CT_, *e*	Zn-Np, Å	<N-Zn-N °
**L1**	−12.97	2.268	38.26	0.103	45.55	0.150	2.074	163.5
**L2**	−19.57	2.183	44.66	0.120	45.07	0.148	2.088	162.9
**L3**	−17.31	2.197	44.47	0.116	43.91	0.144	2.071	163.7
**L4**	−14.93	2.253	38.62	0.101	48.14	0.158	2.074	164.5
**L5**	−12.42	2.256	38.33	0.113	47.50	0.184	2.074	163.2
**L6**	−15.19	2.166	46.18	0.123	46.93	0.150	2.079	163.5

N_p_—nitrogen atom of the pyrrole ring of the macrocycle.

**Table 3 molecules-28-00964-t003:** Geometric, electronic and energy parameters of the CoCP(L1–L6) extra-complexes.

Ligands	E_int,_ kcal·mol^−1^	r(Co-N_Im_), Å	ΣE_st_(LP(N_Im_)→LP*(Co)), kcal·mol^−1^	*q*_st_(LP(N_Im_)→LP*(Co)), *e*	r(Co-N_Pyr_)_,_ Å	ΣE_st_(LP(N_Pyr_)→LP*(Co)), kcal·mol^−1^	*q*_st_(LP(N_Pyr_)→LP*(Co)), *e*	∠N-Co-N, °
**Im**	20.44	2.1184	24.53	0.147	1.9920	34.45	0.143	173.8
**L1**	34.61	2.1992	26.9	0.124	1.9863	54.81	0.143	173.5
**L2**	42.45	2.2232	28.09	0.123	1.9755	56.31	0.143	166.9
**L3**	39.48	2.2355	27.54	0.122	1.9825	55.33	0.141	171.9
**L4**	34.65	2.1498	29.67	0.131	1.9868	54.92	0.142	171.6
**L5**	26.93	2.1584	30.98	0.133	1.9756	49.13	0.149	174.8
**L6**	36.76	2.1366	31.82	0.129	2.0048	58.34	0.152	174.7

**Table 4 molecules-28-00964-t004:** Characteristics of intramolecular hydrogen bonding in CoCP(L).

Ligand	Type	r(H···B), Å	∠A-H···B, °	ΣE_st,_ kcal·mol^−1^
**Im**	-	-	-	-
**L1**	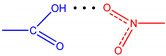	1.83	172.5	13.47
	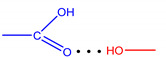	1.98	149.4	
**L2**	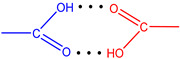	1.75	168.3	22.27
		1.74	156.8	
**L3**	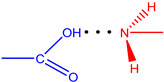	1.67	159.1	36.15
**L4**	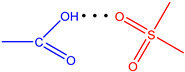	1.75	162.3	5.68
**L5**	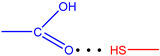	2.24	147.7	3.02
**L6**	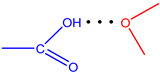	1.85	166.8	6.55

**Table 5 molecules-28-00964-t005:** Proton signals in the ^1^H NMR spectra of the imidazole derivatives in the free form and in the CoCP(L) composition.

L	^1^H NMR-Spectrum of L	^1^H NMR-Spectrum of L in the CoCP(L) Complexes
**L1**	8.027 (s, 1H, Im) 5.3 (br. s, 1H, -OH) 4.36 (t, 2H, -CH_2_-CH_2_-OH) 3.67 (t, 2H, -CH_2_-CH_2_-OH) 2.46 (s, 3H, -CH_3_)	5.86 (br.s, 1H, -OH)-downfield shift 4.23 (t, 2H, -CH_2_-CH_2_OH), 3.48 (t, 2H, -CH_2_-CH_2_OH), 3.28 (s, 1H, 2-Im) -upfield shift 2.28 (s, 3H, -CH_3_).
**L2**	11.92 (s, 1H, -COOH) 8.24 (s, 1H, Im) 7.43 (s, 1H, Im) 6.13 (br. s. 2H, NH_2_) 4.9 (br. s, 1H, NH) 4.06 (s, 1H, -CH-) 3.37 (t, 2H, -CH_2_-)	12.47 (s, 1H, -COOH) 3.43 (s, 1H, Im) 6.32 (s, 1H, Im) 6.24 (br.s, 2H, NH_2_) 4.6 (br. s, 1H, NH) 4.15 (s, 1H, -CH-) 3.31 (t, 2H, -CH_2_-)
**L3**	7.54 (s, 1H, Im) 6.80 (s, 1H, Im) 6.25 (br.s. 2H, NH_2_) 4.98 (br. s, 1H, NH) 2.97 (t, 2H, Im-CH_2_-CH_2_-) 2.75 (t, 2H, Im-CH_2_-CH_2_-)	3.58 (s, 1H, Im) 5.73 (s, 1H, Im) 7,12 (br.s, 2H, NH_2_) 4,90 (br. s, 1H, NH) 2.94 (t, 2H, Im-CH_2_-CH_2_-) 2.70 (t, 2H, Im-CH_2_-CH_2_-)
**L4**	8.027 (s, 1H, Im) 3.48 (s, 2H, -CH_2_-) 3.26 (t, 2H, -CH_2_-CH_2_-CH_3_) 2.16 (t, 2H, -CH_2_-CH_2_-CH_3_) 1.19 (s, 3H, -CH_2_-CH_2_-CH_3_) 2.46 (s, 3H, -CH_3_)	3.61 (s, 1H, Im) 3.32 (s, 2H, -CH_2_-) 4.12 (t, 2H, -CH_2_-CH_2_-CH_3_) 2.16 (t, 2H, -CH_2_-CH_2_-CH_3_) 1.23 (s, 3H, -CH_2_-CH_2_-CH_3_) 2.51 (s, 3H, -CH_3_)
**L5**	12.2 (br. s, 1H, SH) 7.54 (d, 1H, Im) 6.80 (d, 1H, Im) 2.46 (s, 3H, -CH_3_)	12.39 (br.s, 1H, SH) 3.33 (d, 1H, Im) 5.78 (d, 1H, Im), 2.49 (s, 3H, -CH_3_)
**L6**	8.84 (s, 1H, Im) 7.14 (s, 1H, Im) 4.09 (m, 2H, -CH_2_-) 3.66 (m, 2H, -CH_2_-) 3.16 (s, 3H, -CH_3_) 2.98 (s, 1H, -CH<) 2.79 (s, 1H, -CH<) 1.58 (m, 2H, -CH_2_-CH_3_) 1.07 (s, 3H, -CH_2_-CH_3_)	3.38 (s, 1H, Im) 7.03 (s, 1H, Im) 4.02 (m, 2H, -CH_2_-) 3.62 (m, 2H, -CH_2_-) 3.18 (s, 3H, -CH_3_) 3.13 (s, 1H, -CH<) 2.97 (s, 1H, -CH<) 1.52 (m, 2H, -CH_2_-CH_3_) 1.12(s, 3H, -CH_2_-CH_3_)

## Data Availability

Not applicable.

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
