# Peer review of "Molecular Recognition of Imidazole-Based Drug Molecules by Cobalt(III)- and Zinc(II)-Coproporphyrins in Aqueous Media"

_molecules, 2023, doi:10.3390/molecules28030964_

Round 1
Reviewer 1 Report
The work by Mamardashvili et al. investigates the binding of imidazole-containing molecules using metal-containing coproporphyrins. In the introduction it is necessary to justify why zinc and cobalt are taken as the metal ions coordinating the porphyrin ligand. The second question concerns the UV spectra. Can differences between the experimental and calculated absorption spectra be established? This wish follows from the spectra for the cobalt-containing complex with imidazole ligands. The differences in these spectra are minimal, and the conclusion is given about the binding of one or two ligand molecules to the metal centre. Such a conclusion requires stronger evidence.
Changes need to be made in the text of the article and the experimental part. For example, correct d-DMSO to d6-DMSO, DMFA to DMF, etc.
Author Response
Thanks a lot for the valuable comments and recommendations that have significantly improved the article. As a result of the responses, the Supplementary materials file was added to the article.
We are ready to provide additional information if necessary.
With best regards,
the authors.

Reviewer 2 Report
1. Please provide the XPS for confirming the valance state for Co(III).
2. Tables has a better format, why haven't you established a standard?
3. there may be additional binding sites formed due to van der Waals and/or dispersion interactions, hydrogen bonds, Pi-stacking, etc. For example, according to, some work can be highlighted, such as Micropor. Mesopor. Mat, 341(2022) 112098 and Inorganics, 10(2022) 202. Also, IN this part “This can be seen from the values of the N – Zn – N angle, which decreased proportionally to the increase in the energy of the axial ligand binding to the Zn-porphyrin (Table 2).”, This should be also added some refs for comparison, such as RSC Adv., 2017, 7, 10415-10423 and RSC Adv., 2016, 6, 31161-31166
4. Please do the experimental samples for PXRD, which will match with your simulated ones.
5. The manuscript contains spelling/grammatical errors. So, the language should be polished thoroughly. Such as “formation of two OH…О hydrogen bonds in the CoP(L5) complex. and “according to,”6. The authors have stated in the abstract section “It is found that the studied porphyrinates have the highest binding ability towards histamine and histidine, due to the formation of additional hydrogen bonds between the carboxyl groups of the porphyrinate side chains and the binding sites of the ligands.” It should be explained the differences among these complexes in detail.
Author Response

(The authors gave the same response as above.)

Round 2
Reviewer 2 Report
accepted.